# Screening and Elucidation of Chemical Structures of Novel Mammalian α-Glucosidase Inhibitors Targeting Anti-Diabetes Drug from Herbals Used by E De Ethnic Tribe in Vietnam

**DOI:** 10.3390/ph16050756

**Published:** 2023-05-17

**Authors:** Van Bon Nguyen, San-Lang Wang, Tu Quy Phan, Thi Huyen Thoa Pham, Hung-Tse Huang, Chia-Ching Liaw, Anh Dzung Nguyen

**Affiliations:** 1Institute of Biotechnology and Environment, Tay Nguyen University, Buon Ma Thuot 630000, Vietnam; nadzung@ttn.edu.vn; 2Department of Chemistry, Tamkang University, New Taipei City 25137, Taiwan; 3Life Science Development Center, Tamkang University, New Taipei City 25137, Taiwan; 4Department of Science and Technology, Tay Nguyen University, Buon Ma Thuot 630000, Vietnam; phantuquy@ttn.edu.vn (T.Q.P.); pththoa@ttn.edu.vn (T.H.T.P.); 5Division of Chinese Materia Medica Development, National Research Institute of Chinese Medicine, Taipei 11221, Taiwan; kk49310953@nricm.edu.tw (H.-T.H.); liawcc@nricm.edu.tw (C.-C.L.)

**Keywords:** medicinal plants, *Terminalia triptera* Stapf., α-glucosidase inhibitors, (−)-epicatechin, eschweilenol C, Q6P7A9, E De Ethnic Tribe, virtual study

## Abstract

Among ten extracts of indigenous medicinal plants, the MeOH extract of *Terminalia triptera* Stapf. (TTS) showed the most efficient mammalian α-glucosidase inhibition for the first time. The data of screening bioactive parts used indicated that the TTS trunk bark and leaves extracts demonstrated comparable and higher effects compared to acarbose, a commercial anti-diabetic drug, with half-maximal inhibitory concentration (IC50) values of 181, 331, and 309 µg/mL, respectively. Further bioassay-guided purification led to the isolation of three active compounds from the TTS trunk bark extract and identified as (−)-epicatechin (**1**), eschweilenol C (**2**), and gallic acid (**3**). Of these, compounds **1** and **2** were determined as novel and potent mammalian α-glucosidase inhibitors. The virtual study indicated that these compounds bind to α-glucosidase (Q6P7A9) with acceptable RMSD values (1.16–1.56 Å) and good binding energy (DS values in the range of −11.4 to −12.8 kcal/mol) by interacting with various prominent amino acids to generate five and six linkages, respectively. The data of Lipinski’s rule of five and absorption, distribution, metabolism, excretion and toxicity (ADMET)-based pharmacokinetics and pharmacology revealed that these purified compounds possess anti-diabetic drug properties, and the compounds are almost not toxic for human use. Thus, the findings of this work suggested that (−)-epicatechin and eschweilenol C are novel potential mammalian α-glucosidase inhibitor candidates for type 2 diabetes treatment.

## 1. Introduction

Recently, type 2 diabetes (T2D) has been the most popular chronic metabolic disorder demonstrating serious effects and reduction in the life quality of people worldwide [1]. Some therapies, including the use of α-glucosidase inhibitors (aGIs), have been used for the management of T2D [2,3,4]. Several commercial aGIs like acarbose, voglibose, and miglitol are available; however, the therapies using these commercial aGIs may cause some side effects, including flatulence, diarrhea, and abdominal discomfort [5]. Thus, there is a significant demand for the discovery of new natural drugs for the management of T2D.

The approach of using natural products, especially indigenous medicinal plants, is receiving great attention in study and usage for disease treatment and health care for people worldwide [6,7,8]. Nearly 80% of the global population is estimated to rely on plant-derived medicines to address their healthcare needs [9]. The World Health Organization 2011 report also stated that 80% of the population of developing countries used traditional remedies from medicinal plants for the treatment of various diseases, including diabetes. In Vietnam, an estimated 75% percent of the population uses traditional medicine as their primary source of treatment for common health problems [10]. 

Vietnam is a tropical country, currently ranked the sixteenth most biodiverse, with more than 10,000 reported plant species. Of these, around 4000 herbal species have been used as folk medicinal, and about 700 species have been used in traditional medicine [10]. Dak Lak Province is one of five provinces located in the Central Highlands of Vietnam; the area has several national parks and nature reserves, including Yok Don and Chu Yang Sin National Parks and Ea So and Nam Ka Nature Reserves. These locations provide relatively diverse and populous flora. Dak Lak province has a population of around 1.87 million (2019), with 45 ethnic minorities comprising mainly of the E De and M’ Nong ethnic groups [11]. E De ethnic tribes that live in the buffer zones of national parks and nature reserves have valuable indigenous knowledge about traditional remedies from local herbals for the management of various diseases, including cancer, diabetes, and others [5]. 

To date, there have been various studies on the investigation, identification, and listing of plant species in the Central Highlands and of medicinal plants used by the ethnic minorities living in Dak Lak and conservation studies. However, very few studies have been conducted on screening bioactivities, purification, and identification of chemical structures of bioactive compounds from these diverse medicinal plants for the development of drugs or functional foods [12,13,14]. This study aimed to screen, purify, and identify chemical structures of aGIs from medicinal plants used by the E De ethnic tribe for the potential management of T2D. The docking study was also conducted to investigate the interaction and binding energy of the ligands (aGIs) towards the targeting enzyme in this work.

## 2. Results and Discussion

### 2.1. Screening and Evaluation of Medicinal Plants with Potent Inhibition of α-Glucosidase

Among various extracts of medicinal plant samples (Table 1), the MeOH extract of *Terminalia triptera* Stapf. (TTS) demonstrated the efficient inhibition against mammalian α-glucosidase with low half-maximal inhibitory concentration (IC50) values of 178.8 µg/mL compared to other herbal samples (IC50 values greater than 3356 µg/mL) and acarbose, a commercial anti-diabetic drug (IC50 values, 310 µg/mL) (Table 1). Recently, several medicinal plants collected from Dak province were also reported to show anti-α-glucosidase inhibitory activity; these plants included *Euonymus laxiflorus* Champ. trunk bark [5], *Cinnamomum cassia J. S. Presl.* trunk bark [5], *Terminalia bellirica* leaves [5], *Terminalia bellirica* trunk bark [15], *Terminalia corticosa* [15], *Psidium littorale* Raddi leaves, and *Dalbergia tonkinensis* heartwood [12] and their IC50 values for the inhibition of mammalian α-glucosidase were in the range of 360–1720 µg/mL. In this work, we found that the TTS extract showed a lower IC50 value of 178.8 µg/mL; therefore, TTS is suggested as a potential mammalian α-glucosidase inhibitor candidate. 

Though various medicinal plants have been found to possess potential α-glucosidase inhibitory property, α-glucosidase from yeast have been used for the enzyme inhibition assay [13]. However, mammalian α-glucosidases are suggested as the better enzyme source for the evaluation of potent inhibitors because of their similarities with human α-glucosidases [12]. In this study, rat α-glucosidases were conducted for the test. Until now, there have been few reports on the TTS collected in the Central Highland of Vietnam that has been rarely for its rat intestinal α-glucosidase effect. Thus, this result may enrich the valuable effect of TTS for potential use in T2D. 

To explore the most functional part for use, the leaves, trunk bark, and heartwood of TTS were also collected, and their MeOH extracts were tested for anti-α-glucosidase activity (Table 1). Acarbose was also tested for comparison. The extract of heartwood showed weak activity (IC50 ≥ 1000 µg/mL), while the leaves extract showed comparable activity (IC50 = 331 µg/mL) to that of acarbose (IC50 = 309 µg/mL), and the trunk bark extract exhibited the highest activity (IC50 = 181 µg/mL). Thus, this part used was further used for the purification of active compounds. 

### 2.2. Purification and Identification of Active Compounds from Terminalia triptera *Stapf.*

The targeting inhibitors were isolated from TTS based on bioassay-guided purification, and the assessment of the activity of the crude sample fractions, sub-fractions, and purified compounds are presented in Table 2. TTS was first divided into five fractions (named TTS1, TTS2, TTS3, TTS4, and TTS5) via the Diaion HP-20 column. Of these, fraction TTS3 showed the most bioactivity and was chosen for further purification via LH-20 column to obtain seven sub-fractions (TTS31–TTS37). Sub-fraction TTS31 demonstrated the highest enzyme inhibition and was further purified by pre-high-performance liquid chromatography (pre-HPLC), and three compounds were isolated (TTS32-1, TTS32-2, and TTS32-3). The purification process of these compounds was summarized in Figure 1. Before conducting the bioactivity assay and identification of chemical structures, these isolated compounds were confirmed by HPLC for their high purity. All three isolated compounds, TTS32-1, TTS32-2, and TTS32-3, appeared as clear single peaks at the retention time of 5.25 min, 9.0 min, and 9.25 min, respectively, in their HPLC fingerprints (Figure 1), indicating that these isolated compounds had good purity grades and were quailed for chemical identification and testing activity. 

Analysis of the mass, nuclear magnetic resonance (NMR) spectra, and comparison with previously reported constituents revealed that the chemical structures of purified compounds (TTS312-1, TTS312-2, and TTS312-3) were (−)-epicatechin (**1**) [16], eschweilenol C (**2**) (EAR) [17], and gallic acid (GA) (**3**) [18]. The chemical structures of these identified compounds are shown in Figure 1, and their NMR and mass spectra data were recorded and are summarized below: 

**(−)-Epicatechin (1)** (ECC) was obtained as a white amorphous powder; heated electrospray ionization-mass spectrometry (HESI-MS) *m/z*: 289.0720 [M−H]^−^; ^1^H-NMR (500 MHz, MeOH-*d*4): δ 4.84 (^1^H, brs, H-2), δ 4.20 (^1^H, m, H-3), δ 2.88 (^1^H, dd, *J* = 4.5, 16.0 Hz, H-4a), δ 2.75 (^1^H, dd, *J* = 3.0, 16.0 Hz, H-4b), δ 5.96 (^1^H, d, *J* = 2.0 Hz, H-7), δ 5.94 (^1^H, d, *J* = 2.0 Hz, H-9), δ 6.70 (^1^H, d, *J* = 2.0 Hz, H-2′), δ 6.78 (^1^H, d, *J* = 8.0 Hz, H-5′), δ 6.82 (^1^H, ddd, *J* = 1.0, 2.0, 8.0 Hz, H-6′); ^13^C-NMR (125 MHz, MeOH-*d*4): δ 78.5 (C-2), 66.1 (C-3), 27.9 (C-4), 98.7 (C-5), 156.6 (C-6), 95.0 (C-7), 156.3 (C-8), 94.5 (C-9), 156.0 (C-10), 130.9 (C-1′), 113.9 (C-2′), 144.6 (C-3′), 144.4 (C-4′), 114.5 (C-5′), 118.0 (C-6′).

**Eschweilenol C** (**2**) (EWC) was obtained as a yellow amorphous powder; HESI-MS *m/z*: 447.0577 [M−H]^−^; ^1^H-NMR (400 MHz, pyridine-*d5*): δ 8.46 (^1^H, s, H-3), δ 8.11 (^1^H, s, H-3′), δ 6.39 (^1^H, s, H-1″), δ 4.91 (^1^H, brs, H-2″), δ 4.77 (^1^H, brd, *J* = 7.2 Hz, H-3″), δ 4.40 (^1^H, t, *J* = 7.2 Hz, H-4″), δ 4.60 (^1^H, m, H-5″), δ 1.64 (3H, d, *J* = 6.0 Hz, H-6″); ^13^C-NMR (100 MHz, pyridine-*d5*): δ 107.9 (C-1), 115.6 (C-2), 114.6 (C-3), 147.6 (C-4), 144.0 (C-5), 137.7 (C-6), 160.1 (C-7), 108.8 (C-1′), 112.5 (C-2′), 111.6 (C-3′), 150.8 (C-4′), 141.8 (C-5′), 137.8 (C-6′), 160.1 (C-7′), 101.9 (C-1″), 71.7 (C-2″), 72.3 (C-3″), 73.5 (C-4″), 71.2 (C-5″), 18.4 (C-6″). For further confirming this isolated compound (**2**) as eschweilenol C, this isolated compound and a stardar compound of eschweilenol C were analyzed by HPLC. The data in Figure A1 (in the Appendix A) showed that both the stardar compound and compound 2 appeared at the same retention time of approximate 9.8 min. 

**Gallic acid (3)** (GA) was obtained as a white amorphous powder; HESI-MS *m/z*: 169.0130 [M−H]^− 1^H-NMR (400 MHz, MeOH-*d*4): δ 7.05 (2H, s, H-2/H-6); ^13^C-NMR (100 MHz, MeOH-*d*4): δ 122.0 (C-1), 110.3 (C-2), 146.4 (C-3), 139.6 (C-4), 146.4 (C-5), 110.3 (C-6), 170.4 (C-7). To date, various natural compounds have been purified from *Terminalia* species worldwide, and their chemical structures have been elucidated [17,19,20,21,22]. Though several *Terminalia* species growing in the Central Highland of Vietnam are reported to possess some potential medical effects [15], there are only a few studies on the purification and identification of bioactive compounds from these herbal species collected from this region. Notably, this study is the first to report the anti-diabetic effect of herbal TTS. Besides, the bioactive compounds were also isolated and their chemical structures were elucidated. Thus, this study contributes to the novel medical effect of TTS extract and adds to the list of phytochemical compounds of this herbal species. 

### 2.3. Evaluation of the α-Glucosidase Inhibitory Effect of Purified Compounds from Terminalia triptera *Stapf.*

The three compounds, including ECC, EWC, and GA purified in this work, were assessed for their bioactivity against mammalian α-glucosidase. Acarbose was also tested for comparison. As shown in Figure 2, these compounds showed a positive α-glucosidase inhibitory effect. Of these, EWC showed the highest activity with the maximum inhibition (93%) and the lowest IC50 value (191 μg/mL), while ECC showed a comparable effect to that of acarbose with the maximum inhibition and IC50 values in the range of 84–85%, and 304–431 μg/mL, respectively. GA (**3**) demonstrated a weak enzyme inhibition effect, with maximum inhibition and IC50 values of 80%, and 973 μg/mL, respectively.

The isolation and identification of phenolic compounds from various medicinal plants have been widely reported [18,23,24,25,26,27,28]. Of the three phenolics isolated from TTS in this work, ECC and GA are common phenolic compounds, and their bioactivities, including the anti-α-glucosidase inhibition of these compounds, have been often reported [23,27,29]. However, the data concerning the inhibitory effect of these two phenolics against mammalian α-glucosidase are rarely available. In contrast, there are only a few reports on the purification of phenolic compounds EWC from herbs, and this is the first to report on the isolation of this compound from *Terminalia triptera* Stapf. species. According to the literature, EWC was found to be a novel mammalian enzyme-targeting anti-diabetic drug. The current study suggested that TTS extract could be considered a promising anti-diabetic alternative drug due to its containing potent bioactive compounds.

### 2.4. Docking Analysis to Study the Mechanism of Interaction of Purified Compounds with the Target Enzyme, α-Glucosidase 

Recently, docking studies have been widely utilized for investigation of the interaction and binding energy of inhibitors toward targeting enzymes for drug discovery [30,31,32,33,34]. The docking study was performed to elucidate the interaction and the binding energy of the inhibitors toward the target enzyme (mammalian α-glucosidase, Q6P7A9). The protein structure data of the target enzyme was obtained from the Worldwide Protein Data Bank. Then, the most active zones for docking the ligands into were found using the site finder function of the MOE-2015.10 software. Based on the output data of MOE, four binding sites (BSs) were found (Figure 3a), and the amino acid residues in these BSs were also determined and presented in the Appendix A (Table A1). The output data of MOE show that all of the three purified active compounds ECC, EWC, and GA) interacted the most effectively with the enzyme at binding sites 1 (BS1), while acarbose (AC) bound effectively to the enzyme at binding sites 4 (BS4).

To determine whether or not these BS of ligands are covered by the catalytic site (active site) of enzyme, we utilized the CASTp3.0 server for prediction of the catalytic site of Q6P7A9. The 3D structure of the catalytic site was shown in Figure 3b, and the detailed data of this catalytic site of Q6P7A9 are presented in the Appendix A (Table A2). The volume and the surface area of this active site were found to be 15,668.797 Å^3^ and 4922.961 Å^2^, respectively. As shown in Figure 3, three binding sites (BS1, BS2, and BCS4) were located in the catalytic site. Since the three compounds (ECC, EWC, and GA) isolated in this work and acabose were found most effectively with the enzyme at BS1 and BS4, respectively, as such, all these inhibitors show high possibility binding to the enzyme Q6P7A9 at the catalytic site.

In docking performance, the root mean square deviation (RMSD) and docking score (DS) are commonly considered important values for determining the successful interaction and potent inhibition of a ligand (inhibitor) towards a target protein (enzyme). When a ligand interacts with its targeting enzyme with RMSD and DS values of less than 2.0 Å and less than −3.20 kcal/mol, respectively, it should be proposed as an effective inhibitor with significant interaction and binding energy toward the targeting enzyme [36,37]. As summarized in Table 2, all the examined four ligands, including ECC, EWC, GA, and acarbose, successfully interacted with mammalian α-glucosidase (Q6P7A9) with low RMSD values, respectively, of 1.56, 1.16, 0.77, and 0.94. In addition, the DS values of these four ligands interacting with α-glucosidase were in the range of −10.1 (kcal/mol) to −12.8 kcal/mol), indicating that they are potent enzyme inhibitors. Based on the comparison of binding energy (DS), EWC was the strongest inhibitor [DS = −12.8 (kcal/mol)], while that of ECC and acarbose were in the range of −11,4 and −11.4 kcal/mol, respectively, and the DS value of GA, when bound to the enzyme, was the highest at −10.1 kcal/mol. These virtual data seem in agreement with the experimental data shown in Figure 2.

To inside understand the interaction of ligands and enzyme, the detailed binding at the BSs were recorded and are presented in Figure 4 (2D structures) and Figure A2, Figure A3, Figure A4 and Figure A5 (3D structures, in the Appendix A). Of these, ligand (−)-epicatechin was bound to BS1 on mammalian α-glucosidase (Q6P7A9) via interaction with some prominent amino acids, including ASP282, MET519, MET519, ASP616, and ARG600, resulting in the formation of five bonds. ASP282 was found to form 2 H-donor linkages via interaction with O 14 of (−)-epicatechin, and MET519 also created one H-donor linkage by linking to O 14 of (−)-epicatechin. ASP616 and ARG600 interacted, respectively, with O 20 and O 14 to form one H-donor and one H-acceptor linkage. The distance and energy of these linkages are recorded in Table 1.

The ligand EWC was found to bind to BS1 on Q6P7A9 by six linkages by interacting with some prominent amino acids. ASP518 interacted with C 26 and O 32 of ligand EWC for the creation of 2 H-donor linkages. ASP404 contacted with O 30 and O 31 of ligand EWC, and 2 H-donor linkages were also formed. ASP282 and ALA284 were found to bind to O 28 and O 20 for creating two linkages, respectively, of H-donor and H-acceptor. The binding distance and energy of these linkages are also recorded in Table 1. Ligand GA could also bind to Q6P7A9 at the BS1 via interaction with three prominent amino acids, MET519, ARG600, and TRP613 at O 7, O 11, and O12, respectively, to form three linkages, including 1 H-donor, 2 H-acceptors. Acarbose, a commercially available inhibitor, was also assessed in the virtual study. This ligand was found to affect the binding to BS1 by interacting with some amino acids and formed 5 linkages (2 H-donor, 3 H-acceptors). The distance and binding energy of these linkages of these ligands towards the targeting enzyme were also recorded and are summarized in Table 1.

Being common phenolics compounds, the bioactivities of ECC and GA were widely investigated via experimental studies and virtual performance. The docking study to inside understand the interaction and energy binding of these two compounds have been performed for some sources of α-glucosidase; however, the docking analysis of the interaction of these three ligands with mammalian α-glucosidase has been rarely reported. Notably, this is the first report on the interaction inside the cavity and the binding energy of EWC at the most active zone on the target enzyme, α-glucosidase. According to the experimental analysis and the virtual evaluation, the isolated compounds, including ECC and EWC, are suggested as potential candidates for the management of T2D. However, further studies in various animal models and clinical studies are needed for the development of these potential inhibitors into anti-diabetes drugs.

### 2.5. The Lipinski’s Rule of Five and Absorption, Distribution, Metabolism, Excretion, and Toxicity (ADMET)-Based Pharmacokinetics and Pharmacology

The Lipinski’s rule of five has been widely applied to determine whether a compound processing drug properties according to the five rules, including: “molecular mass must be less than 500 Da, high lipophilicity with LogP value < 5, hydrogen bond donors < 5, hydrogen bond acceptors < 10, and the molar refractivity should be between 40–130”. A high probability of success to drug-likeness for compounds complying with more than two of Lipinski’s rule of five can thus be predicted [38]. According to the results presented in Table 3, ECC, EWC, and GA complied with five, three, and Lipinski’s rule of five; as such, these inhibitor compounds have a high probability of successfully being developed as a drug. Acarbose, a commercial anti-diabetic drug, was also analyzed by these rules; however, this commercial compound only satisfied approximately two rules.

The ADMET properties of the compounds purified from TTS were also accessed and presented in Table 4. Concerning the absorption ability of a drug, a compound is considered a poor absorption agent if the ADMET value is recorded under 30%. As summarized in Table 4, all these compounds showed a high level of intestinal absorption (human) values in the range of 40.154–72.619%, while acarbose showed a low absorption value of 4.172%. Considering drug distribution, ECC and EWC possessed the log VDss values of 0.215 and 0.53 log L.kg^−1^, respectively, indicating that these inhibitors show a plasma–tissue balance (−015 < log VDss < 0.45), while GA and acarbose were not able to cross the blood–brain barrier (BBB) due to their low BBB permeability values (logBB  <  − 1). The three compounds purified in this work showed a slight effect on the central nervous system (logPS in the range of −3.403 to −4.501), while acarbose, with very low CNS permeability (logPS  <  − 6.438), showed a significant effect on the central nervous system. Concerning metabolism, all these examined compounds displayed no effects on the inhibitors and substrates related to the cytochromes P450 family, indicating that these molecules may not be oxidized by the liver and have a longer stay in the living system [39]. Regarding excretion, the prediction of all the compounds was carried out via the Organic Cation Transporter 2, with their total clearance values in the range of 0.2–1.5 log mL.min^−1^.kg^−1^. In terms of toxicity, the three inhibitors (EWC, GA, and AC) were identified as safe molecules for human use with no mutagenic potentials (AMES toxicity), no potential for fatal ventricular arrhythmia (hERG inhibition), and no potential for hepatotoxicity as well as no toxicity to skin sensitisation. Only, the compound **1** (ECC) showed its positive result to AMES toxicity, while it showed no positive result with other toxicity tests, including hERG I inhibitor, hERG II inhibitor, hepatotoxicity, and skin sensitisation.

## 3. Materials and Methods

### 3.1. Materials

The samples of medicinal plants were collected from the Central Highland of Vietnam in 2018. The name and parts used by these herbals are mentioned in Table 1. The dried plant parts were then packed in PE bags and stored at −30 °C before use. Acarbose was purchased from Sigma Chemical Co. (St. Louis, MO, USA). Rat intestinal acetone powders (rat α-glucosidase) were purchased from Sigma Aldrich, Singapore. *p*-Nitrophenyl glucopyranoside (*P*npg) was purchased from Sigma Aldrich (3050 Spruce Street, St. Louis, MO, USA). Diaion HP-20 was purchased from the Mitsubishi Chemical Co. (Tokyo, Japan). The solvents, reagents, and other commonly used chemicals were of the highest grade available.

### 3.2. General Process for Purification and Elucidation of Chemical Structures of Inhibitors Compounds

Fifty-five grams of methanol extract of TTS were separated via the Diaion HP-20 column by eluting with MeOH in water at 15%, 30%, 45%, 60%, and 100% gradient to obtain five fractions named TTS1, TTS2, TTS3, TTS4, and TTS5, respectively. TTS3 was chosen for further separation by LH-20 column and eluted with MeOH in water at 70–100% gradient, and seven sub-fractions were obtained (named from TTS31–TTS37). The application of pre-HPLC for TTS32 and elution at 10 MeOH yielded three compounds named TTS32-1, TTS32-2, and TTS32-3. The separation and purification process of these compounds from TA extract are summarized in Figure 1. The chemical structures of purified compounds were elucidated after analyzing their masses, NMR spectra, and comparison with the previously reported constituents. The ^1^H and 13C-NMR spectra and the 2D-NMR spectra were measured in MeOH-*d*4 on a Bruker AVX NMR spectrometer (Bruker, Karlsruhe, Germany), operated at 600 MHz for 1–12 h and 150 MHz for 13C spectra measurement, and the MeOH-*d*4 solvent peak was used as the internal standard (δH 3.317, δC 49.1 ppm).

### 3.3. High-Performance Liquid Chromatography Analysis

The compounds were dissolved in MeOH at 1 mg/mL and filtered using a 0.45 μm PVDF membrane filter (Millipore Sigma, Billerica, MA, USA) before use. Ten microliters of the compound solution were injected into the column (Cosmosil 5C_18_-AR-II, 5 μm, 250 × 4.6 mm i.d.) and separated by HPLC (SHIMADZU SIL-40C) at 45 °C with a mobile phase of acetonitrile (ACN) and water containing 0.3% phosphoric acid. The mobile phase program was set at 5–5–15–45–100% ACN from 0–10–20–40–50 min, respectively. The flow rate was set at 1 mL/min, and the compounds were detected at 210 nm.

### 3.4. Alpha-Glucosidase Inhibition Assay

-Enzyme solution preparation: Enzyme solution was then sonicated 24 times, each for 12 s at 4 °C, and then centrifuged for 20 min at 10,000× *g*, 4 °C. The residue part was re-suspended twice with 15 mL of the same buffer (0.1 mol/L NPB, pH 7), as described above. All supernatants were mixed together and dialyzed for a half day at 4–6 °C to obtain the enzyme solution, which was then used for further analyses.-Reaction and estimation of inhibitory activity: The enzyme inhibition assay was done according to Kwon et al. 2006 [40] with slight modification. The enzyme solution (50 µL) and the sample solution (50 µL) were mixed in 150 μL of 0.1 mol/L NPB, pH 7. This mixture was pre-incubated at 37 °C for 15 min. The reaction was started by adding 50 μL of the substrate pNPG (10 mmol/L) into this mixture solution, and this duration was kept at 37 °C for 30 min, then 325 μL of 1 mol/L Na_2_CO_3_ solution were added to terminate the reaction. This absorption of the final solution was measured at 410 nm (namely, E) using a UV spectrometer (UV-2550, Shimadzu, Japan) to estimate the enzymatic activity, and the control experiment was also conducted similarly, but instead of the inhibitor solution, 50 μL 0.1 mol/L potassium phosphate buffer (pH 7) and the absorbance was measured at 410 nm (namely, C). The inhibitory effect (%) was calculated using the equation below:

α-Glucosidase inhibition (%) = (C − E)/C × 100,
where E denoted the optical density of the solution reaction in the presence of both α-glucosidase and the sample, and C is the optical density of the reaction blank. The IC50 value was defined as the concentration of an enzyme inhibitor required to inhibit 50% of the α-glucosidase activity under the assay conditions.

### 3.5. Computation

#### 3.5.1. Molecular Docking Simulation

The virtual simulations were done following some typical steps previously presented in the earlier reports [41,42,43,44,45]:

Protein (α-glucosidase) structure preparation and finding the active site on an enzyme: The data structure of protein enzyme α-glucosidase was obtained from the Worldwide Protein Data Bank to prepare its 3-D structure using the software MOE-2015.10. The virtual pH was set at 7 for the preparation of the α-glucosidase structure. The function of the site finder in MOE-2015.10 software was used for finding the most active zone on the 3-D α-glucosidase structure.

Ligands (enzyme inhibitors) preparation: The structure of three inhibitor ligands and acarbose were prepared using ChemBioOffice 2018 software, then their structures were further optimized using MOE-2015.10 software. The parameters included force field MMFF94x; R-field 1:80; cutoff, rigid water molecules, space group p1, cell size 10, 10, 10; cell shape 90, 90, 90; gradient of 0.01 RMS kcal.mol^−1^A^−2^; virtual pH 7 was set for preparing the structures of the ligands.

Docking and harvesting the output data: The ligands were docked into the active zone of the targeting protein using MOE-2015.10 software, and some output data, including DS, RMSD, linkage types, compositions of amino acids, and the linkages distances, were obtained for analysis.

#### 3.5.2. The Lipinski’s Rule of Five and Absorption, Distribution, Metabolism, Excretion, and Toxicity Analysis Protocol

A virtual study to investigate the Lipinski’s rule of five was performed using the software online accessed at (http://www.scfbioiitd.res.in/software/drugdesign/lipinski.jsp; accessed on 3 March 2023). Some pharmacokinetic parameters, including ADMET, were obtained for analysis via using a web tool SwissADME (http://www.swissadme.ch/, accessed on 18 April 2023). The output data of theoretical interpretations of pharmacokinetic parameters have been previously described [46] and used as public reference online accessed at (http://biosig.unimelb.edu.au/pkcsm/theory; accessed on 18 April 2023).

## 4. Conclusions

Among various sample extracts of indigenous medicinal plants, the MeOH extract of TTS showed the most efficient inhibition against mammalian α-glucosidase. The targeting inhibitor compounds were isolated from TTS based on bioassay-guided purification and identified as ECC, EWC, and GA. In the activity evaluation, EAR demonstrated significant α-glucosidase inhibition—reported for the first time—and this effect is higher than acarbose, a commercially available anti-diabetic compound, while ECC showed comparable activity to that of acarbose. Of the purified compounds, GA was less effective against α-glucosidase. The docking study was also conducted to elucidate the interaction and energy binding of these inhibitor compounds toward the targeting enzyme. The data of this work contribute to the reclamation of novel in vitro anti-diabetic effects of TTS and enrich the list of phytochemical compounds from this herbal species. Based on the analysis of the Lipinski’s rule of five and ADMET-based pharmacokinetics and pharmacology, these inhibitors were found to possess drug properties, and almost all of these inhibitors exhibited no toxicity. The experimental data and the virtual evaluation showed that the two isolated compounds, including (−)-epicatechin and EWC, are potential candidates for T2D treatment. However, more studies concerning the evaluation of the effect and toxicity of these candidates in various animal models and clinical studies should be further conducted for the assessment and development of these potential inhibitors into anti-diabetes drugs.

## Data Availability

Not applicable.

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
