# Peer review of "Screening and Elucidation of Chemical Structures of Novel Mammalian α-Glucosidase Inhibitors Targeting Anti-Diabetes Drug from Herbals Used by E De Ethnic Tribe in Vietnam"

_pharmaceuticals, 2023, doi:10.3390/ph16050756_

Round 1
Reviewer 1 Report
The anti-diabets constituents in the Vietnam herbal of Terminalia triptera Stapf. (TTS) have been studied in this manuscript. The crude extracts and the purified compounds are evaluated for the α-glucosidase inhibition activity. These results showed some significance for treating type-2 diabetes. However, the current manuscript still need some refinements on data presenting and especially on elucidating the structures of the obtained compounds.
1. Structures of pure compounds should be presented with carbon numbers. And please make the carbon numbers in consistent with those from the references cited in the body text. This may help check the chemical shifts these compounds with the compounds already reported.
2. Please use the same name as shown in the cited references. For example, the compounds named “Ellagic acid 4-O-α-L-rhamnopyranoside” was named eschweilenol C in reference 17.
3. The chemical shifts of compound 2 showed significant differences compared with the data reported in the literature, the authors are encouraged to use 2D NMR data and assisted by hydrolysis of the sugar part to exactly determine the absolute configuration of this compound. Or else, the author may provide a standard of Ellagic acid 4-O-α-L-rhamnopyranoside to identify compound 2 by HPLC.
Author Response
Dear Reviewer/Advancer
We feel pleasure to thank you for all of your time and effort, as well as your excellent suggestions for refining the readability and impact of our manuscript. We have gone through all the suggestions cautiously and made the revisions accordingly; and all revised parts have been typed in red in the revised manuscript. Finally, we would like to express our deep thanks to your comments and suggestions again. You certainly have served to improve the quality of this paper. We hope our response may meet the satisfactory.
Looking forward to hearing from you.
Thanking you,
Yours Sincerely,
Reviewer 1:
Comments and Suggestions for Authors
The anti-diabets constituents in the Vietnam herbal of Terminalia triptera Stapf. (TTS) have been studied in this manuscript. The crude extracts and the purified compounds are evaluated for the α-glucosidase inhibition activity. These results showed some significance for treating type-2 diabetes. However, the current manuscript still need some refinements on data presenting and especially on elucidating the structures of the obtained compounds.
- Structures of pure compounds should be presented with carbon numbers. And please make the carbon numbers in consistent with those from the references cited in the body text. This may help check the chemical shifts these compounds with the compounds already reported.
Reply: Thanks so much for your very positive comments and have interesting view on our work. We have added the chemical structures and assign the carbon numbers in Figure 1.
- Please use the same name as shown in the cited references. For example, the compounds named “Ellagic acid 4-O-α-L-rhamnopyranoside” was named eschweilenol C in reference 17.
Reply: We revised “Ellagic acid 4-O-α-L-rhamnopyranoside” to “eschweilenol C” in the manuscript accordingly your comment.
- The chemical shifts of compound 2 showed significant differences compared with the data reported in the literature, the authors are encouraged to use 2D NMR data and assisted by hydrolysis of the sugar part to exactly determine the absolute configuration of this compound. Or else, the author may provide a standard of Ellagic acid 4-O-α-L-rhamnopyranoside to identify compound 2 by HPLC.
Reply: For further confirm according to your suggestion, We used the standard of eschweilenol C and compared with compound 2, the retention time (9.835 and 9.810 min) is the same (Shown in the Appendix A1). One more time, we would like to pay our deeply thanks for your very careful review, and valuable comments for the enhancement of the quality of the manuscript.

Reviewer 2 Report
The manuscript entitled "Screening and Elucidation of Chemical Structures of Novel Mammalian α-Glucosidase Inhibitors Targeting..." has been evaluated. The manuscript presents the novelty of 2 common compounds against rat intestinal alpha-glucosidase. However, I found that some experiments may lead to misleading results.
1) The substrate should be disaccharides such as maltose or sucrose. The use of PNPG may result in error in detecting the absorption at 410 nm, which is easily interfered with by impurities, especially in the screening step of crude extracts. The use of glucose oxidase kit can guarantee the absorption detected at 575 nm and can make sure the prevention of interference from crude extract color.
2) Prior to applying molecular docking to determine the binding site of your inhibitors, the authors need to do a kinetic study and determine the mode of inhibitory (e.g. competitive), otherwise the author do not know the inhibitor bind to the enzyme at the active site or binding site.
3) In table 1, the plant codes such as TTS refered to Terminalia triptera Stapf. should be added to facilitate the readers easily to understand.
Author Response
Dear Reviewer/Advancer
We feel pleasure to thank you for all of your time and effort, as well as your excellent suggestions for refining the readability and impact of our manuscript. We have gone through all the suggestions cautiously and made the revisions accordingly; and all revised parts have been typed in red in the revised manuscript. Finally, we would like to express our deep thanks to your comments and suggestions again. You certainly have served to improve the quality of this paper. We hope our response may meet the satisfactory.
Looking forward to hearing from you.
Thanking you,
Yours Sincerely,
Reviewer 2:
Comments and Suggestions for Authors
The manuscript entitled "Screening and Elucidation of Chemical Structures of Novel Mammalian α-Glucosidase Inhibitors Targeting..." has been evaluated. The manuscript presents the novelty of 2 common compounds against rat intestinal alpha-glucosidase. However, I found that some experiments may lead to misleading results.
1) The substrate should be disaccharides such as maltose or sucrose. The use of PNPG may result in error in detecting the absorption at 410 nm, which is easily interfered with by impurities, especially in the screening step of crude extracts. The use of glucose oxidase kit can guarantee the absorption detected at 575 nm and can make sure the prevention of interference from crude extract color.
2) Prior to applying molecular docking to determine the binding site of your inhibitors, the authors need to do a kinetic study and determine the mode of inhibitory (e.g. competitive), otherwise the author do not know the inhibitor bind to the enzyme at the active site or binding site.
Reply 1-2: We already conducted the study using the assay which has been wildly utilized for a-glucosidase inhibition assay reported in many previous reports [A, B, C, D, ect.,..]. As you understanding, its often not easy to obataining a large amount of natural compounds isolated from medicinal plants. Almost the purified compounds were used for Identification of structures (NMR, Mass) and bioassay tests. Thus, we are unable to re-test the activity according to this comment, and thus, may not more tests to find the kinetics.
In enzyme docking studies, inhibitor compounds may interact and bind to the target enzyme proteins at numerous binding sites (BSs). Thus, normally, only the one BS possessing the lowest binding energy is chosen for description in detail for each inhibitor, and we already followed and conducted the docking study with some typical steps according to the protocol recently accepted in various previous publications [D, E, F, J, etc]. In this docking study, we already provide the optimized 3D Structure of enzyme, and the binding site (active zone of each ligands “inhibitor”) was determined by using site finder function of MOE. The exact data (prominent amino acids containing in the binding site) and the 3 D structure of binding sites on the targeting protein enzyme a-glucosidase of these purified compounds (1-3) were clearly indicated. Then the detail values of virtual study, including RMSD, DS, linkages as well as amino acids interacting with the ligands were presented and discussed, similar to those of numerous studies being recently published [D, E, F, G, H, etc,..]. However, your comment is also valuable to inside understand the interaction of ligands (inhibitor) and target protein (enzyme), as such, we already carefully check and read more publication concerning virtual study. We used CASTp3.0 server [L] for finding the active site (catalytic site) of enzyme (for prediction of catalytic site of α-glucosidase (Q6P7A9) and compare ring the data of active site and binding sites to determine the ligands interacting with enzyme via active site or binding sites. This information was addressed in the revised version now. Thanks for your direction comments.
Refferences (Publications in Web of Sciences):
[A]. https://link.springer.com/article/10.1007/s11164-019-04019-4;
[B]. https://linkinghub.elsevier.com/retrieve/pii/S1359511318312133
[C]. https://linkinghub.elsevier.com/retrieve/pii/S1359511317310413;
[D]. https://www.mdpi.com/1420-3049/26/20/6270
[E]. https://www.mdpi.com/1660-3397/20/5/283
[F]. https://doi.org/10.1039/D1RA00441G
[G]. https://doi.org/10.1039/D1RA04461C
[H]. https://doi.org/10.3390/pr11030880
[I]. https://doi.org/10.1007/s11696-022-02273-2
[J]. https://doi.org/10.3390/plants11030388
[K]. https://doi.org/10.9734/bpi/ctcb/v7/17662D
[L]. https://doi.org/10.3390/molecules26144147
3) In table 1, the plant codes such as TTS refered to Terminalia triptera Stapf. should be added to facilitate the readers easily to understand.
Reply 3: We added the plant codes in a column of Table 1. One more time, we would like to pay our deeply thanks for your very careful review, and valuable comments for the enhancement of the quality of the manuscript as well as our future study designation.
Reviewer 3 Report
In this paper Van Bon Nguyen et al. report the evaluation of the potential antidiabetic effects of native plants from the Central Highland of Vietnam. The rate of type 2 diabetes is increasing, therefore the search for new antidiabetic drugs is important.
The authors provided valuable results about the alpha-glucosidase inhibitory effect of Terminalia Triptera Stapf extracts, and also isolated and identified the active compounds responsible of the effect.
I have some minor questions:
1. Leaf extract of TTS showed comparable effect to that of acarbose, while bark extract surpassed that. That's why in the paper only the bark extract was used for isolation. Do the authors know if the leaves have the same active compounds responsible for the inhibitory effect, just in lower concentration?
2. In table 1. only the bark extract of TTS is shown, but in the text the leaf and heartwood extract is mentioned as well. In case of the other plants only the extracts mentioned in the table were tested, or other parts of the plants too? If so, than that should be mentioned somewhere in the text (e.g. the experimental section). If not, than the table can be supplemented with the other two extracts of TTS.
3. The text at line 419 says that none of the compounds showed toxicity (AMES), but in table 4. "yes" is written for epicatechin AMES toxicity.
Language of the manuscript needs thorough revision. The correct name is Lipinski's rule of five.
Overall after answering my questions and revising the English of the manuscript I recommend publication.
Author Response
Dear Reviewer/Advancer
We feel pleasure to thank you for all of your time and effort, as well as your excellent suggestions for refining the readability and impact of our manuscript. We have gone through all the suggestions cautiously and made the revisions accordingly; and all revised parts have been typed in red in the revised manuscript. Finally, we would like to express our deep thanks to your comments and suggestions again. You certainly have served to improve the quality of this paper. We hope our response may meet the satisfactory.
Looking forward to hearing from you.
Yours Sincerely,
Reviewer 3:
Comments and Suggestions for Authors
In this paper Van Bon Nguyen et al. report the evaluation of the potential antidiabetic effects of native plants from the Central Highland of Vietnam. The rate of type 2 diabetes is increasing, therefore the search for new antidiabetic drugs is important.
The authors provided valuable results about the alpha-glucosidase inhibitory effect of Terminalia Triptera Stapf extracts, and also isolated and identified the active compounds responsible of the effect.
Reply: Thanks so much for your very positive comments and have interesting view on our work.
I have some minor questions:
- Leaf extract of TTS showed comparable effect to that of acarbose, while bark extract surpassed that. That's why in the paper only the bark extract was used for isolation. Do the authors know if the leaves have the same active compounds responsible for the inhibitory effect, just in lower concentration?
Reply: Thanks for your suggestion comment. At present in our manuscript, various natural compounds have been purified from Terminalia species worldwide, and their chemical structures have been elucidated. Though several Terminalia species growing in the Central Highland of Vietnam are reported to possess some potential medical effects, there are few available data of phytochemical compounds were isolated and identified form Terminalia triptera Stapf. Thus, at the moment, we do not know concerning this question. This question should be considered for further designed studies concerning chemical profiles and medical effects of different parts used and different solvents extracts, as well as the study on qualified some biomarkers compounds of this herbal species.
- In table 1. only the bark extract of TTS is shown, but in the text the leaf and heartwood extract is mentioned as well. In case of the other plants only the extracts mentioned in the table were tested, or other parts of the plants too? If so, than that should be mentioned somewhere in the text (e.g. the experimental section). If not, than the table can be supplemented with the other two extracts of TTS.
Reply: The data of other extracts of TTS, including the leaf and heartwood extracts were added in Table 1 accordingly the comment, thank you for pointing this item.
- The text at line 419 says that none of the compounds showed toxicity (AMES), but in table 4. "yes" is written for epicatechin AMES toxicity.
Reply: I am sorry for the mistake, this Item was revised. Thank you.
Language of the manuscript needs thorough revision. The correct name is Lipinski's rule of five.
Overall after answering my questions and revising the English of the manuscript I recommend publication.
Reply: We already checked and revised accordingly the comment. The Manuscript was also checked by a native English speaker before our submission. One more time, we would like to pay our deeply thanks for your very careful review, and valuable comments for the enhancement of the quality of the manuscript as well as our future study designation.
Round 2
Reviewer 1 Report
The quality of figures and charts could be improved.